Genome-wide identification of 2-oxoglutarate and Fe (II)-dependent dioxygenase family genes and their expression profiling under drought and salt stress in potato

http://orcid.org/0009-0004-0939-5157 Chauhan Hanny
Aiana
http://orcid.org/0000-0002-9646-9043 Singh Kashmir kashmirbio@pu.ac.in
Department of Biotechnology, Panjab University , Chandigarh , India
Irfan Mohammad
Electronic publication date: 2023 Nov 20
Publication date: 2023
Volume: 11
Electronic Location ID: e16449
Received 2023 Apr 28; Accepted 2023 Oct 23
Copyright: © 2023 Chauhan et al.
Copyright year: 2023
Copyright holder: Chauhan et al.
License: This is an open access article distributed under the terms of the Creative Commons Attribution License, which permits unrestricted use, distribution, reproduction and adaptation in any medium and for any purpose provided that it is properly attributed. For attribution, the original author(s), title, publication source (PeerJ) and either DOI or URL of the article must be cited.
License URL: https://creativecommons.org/licenses/by/4.0/

Keywords: 2ODDs, Potato, Abiotic-stress, Secondary metabolites, Differential expression analysis

Funding: Department of Biotechnology (DBT) Government of India who provided the resources and financial support for the research Hanny Chauhan received junior and senior research fellowships from the Council of Scientific and Industrial Research (CSIR) Aiana Gill received junior and senior research fellowships from the University Grants Commission (UGC), New Delhi This work was supported by the Department of Biotechnology (DBT), Government of India who provided the resources and financial support for the research. Hanny Chauhan received junior and senior research fellowships from the Council of Scientific and Industrial Research (CSIR). Aiana Gill received junior and senior research fellowships from the University Grants Commission (UGC), New Delhi. The funders had no role in study design, data collection and analysis, decision to publish, or preparation of the manuscript.

==============================
The 2-Oxoglutatrate-dependent dioxygenases (2OGDs) comprise the 2-Oxoglutatrate and Fe(II)-dependent dioxygenases (2ODD) enzyme families that facilitate the biosynthesis of various compounds like gibberellin, ethylene, etc. The 2OGDs are also involved in various catabolism pathways, such as auxin and salicylic acid catabolism. Despite their important roles, 2ODDs have not been studied in potato, which is the third most important crop globally. In this study, a comprehensive genome wide analysis was done to identify all 2ODDs in potatoes, and the putative genes were analysed for the presence of the signature 2OG-FeII_Oxy (PF03171) domain and the conserved DIOX_N (PF14226) domain. A total of 205 St2ODDs were identified and classified into eight groups based on their function. The physiochemical properties, gene structures, and motifs were analysed, and gene duplication events were also searched for St2ODDs. The active amino acid residues responsible for binding with 2-oxoglutarate and Fe (II) were conserved throughout the St2ODDs. The three-dimensional (3D) structures of the representative members of flavanol synthase (FNS), 1-aminocyclopropane-1-carboxylic acid oxidases (ACOs), and gibberellin oxidases (GAOXs) were made and docked with their respective substrates, and the potential interactions were visualised. The expression patterns of the St2ODDs under abiotic stressors such as heat, salt, and drought were also analysed. We found altered expression levels of St2ODDs under abiotic stress conditions, which was further confirmed for drought and salt stress using qRT-PCR. The expression levels of St2ODD115, St2ODD34, and St2ODD99 were found to be upregulated in drought stress with 2.2, 1.8, and 2.6 fold changes, respectively. After rewatering, the expression levels were normal. In salt stress, the expression levels of St2ODD151, St2ODD76, St2ODD91, and St2ODD34 were found to be upregulated after 24 hours (h), 48 hours (h), 72 hours (h), and 96 hours (h). Altogether, the elevated expression levels suggest the importance of St2ODDs under abiotic stresses, i.e., drought and salt. Overall, our study provided a knowledge base for the 2ODD gene family in potato, which can be used further to study the important roles of 2ODDs in potato plants.

Introduction

The iron-containing, non-heme enzymes known as 2-Oxoglutarate-dependent dioxygenases (2OGDs) localise in the cytosol and convert 2-Oxoglutarate to succinate and carbon dioxide. The 2OGDs are ubiquitously distributed in nature, ranging from bacteria, fungi, and plants to vertebrates and parts in various oxygenation and hydroxylation reactions (de Carolis & de Luca 1994; Cheng et al., 2014; Jiang et al., 2021). The 2OGDs superfamily comprises the largest enzyme family in the plant genome after cytochrome P450 monooxygenases (CYPs). The amino acid sequences of 2OGDs differ greatly, and they are involved in various metabolic pathways. Proline hydroxylase was the first identified 2OGD, which requires α-ketoglutarate, ascorbate for binding, and a ferrous ion as a cofactor for its activation (Hutton & Udenfriend, 1966; Ge et al., 2021). Earlier genome exploration of six model plant species classified 2OGDs into three broad classes: DOXA, DOXB, and DOXC, based on their amino acid sequence similarity (Kawai, Ono & Mizutani, 2014; Wang et al., 2022). Among the three classes of 2OGD proteins, DOXA is composed of Escherichia coli AlkB plant homologs, which undergo oxidative demethylation of alkylated nucleic acids and histones (Falnes, Johansen & Seeberg, 2002; Trewick et al., 2002). The DOXB class undergoes hydroxylation of proline in cell wall protein synthesis (Keskiaho et al., 2007; Hieta & Myllyharju, 2002). The DOXC class is the most significant class of 2OGDs as it plays a role in plant metabolism and is involved in various pathways, including steroidal glycoalkaloids (SGA) and flavonoid biosynthesis (Hagel & Facchini, 2018; Sonawane et al., 2022). The 2-Oxoglutatrate and Fe(II)-dependent dioxygenases (2ODD) gene families are members of the DOXC class. The 2ODDs have the signature 2OG-FeII_Oxy (PF03171) domain and also have the conserved DIOX_N (PF14226) domain (Kawai, Ono & Mizutani, 2014). The 2ODDs play a role in the secondary metabolism of plants. Additionally, they play significant roles in a number of biosynthesis and catabolism processes, such as the metabolism of SGA, ethylene, auxin, gibberellin, and salicylic acid (Farrow et al., 2014; Sonawane et al., 2022). The versatility of 2ODD enzymes in various biosynthetic pathways for important metabolite synthesis and normal plant functioning makes the study of the 2ODD gene family important (Pan et al., 2017).

Potato (Solanum tuberosum) is an important cash crop and the world’s third-most important crop. It is globally consumed and is a rich source of carbohydrates and vitamins. It is also used to make commercial food products (You et al., 2019; Alok et al., 2022; Zaki & Radwan, 2022). Potatoes are also linked to a number of anti-nutritional substances, such as α-solanine and α-chaconine which belong to St2ODDs. Plant secondary metabolites α-solanine and α-chaconine are toxic to humans at quantities of 200 to 400 mg/kg (Machado, Toledo & Garcia, 2007; Liu et al., 2021). Prior studies have targeted these secondary metabolites and reported reducing the concentration of these antinutritional compounds through metabolic profiling (Nakayasu et al., 2018). In addition, potato yield is affected by various abiotic stresses like heat, drought, and salt, and therefore, it becomes important to identify stress-responsive St2ODDs affecting the secondary metabolites. Tuber yield is of much interest in order to develop new crop varieties resistant to elevated abiotic stresses in nature.

In potatoes, the 2ODD gene family has not been characterised. In this study, St2ODDs belonging to the DOXC class were studied, and 205 St2ODD genes were identified. Further, these identified genes were systematically analysed for their gene structure, conserved motifs, physiological properties, evolutionary relationship, chromosomal distribution, duplication events, i.e., tandem duplication, and segmental duplication. Active sites were also predicted for substrate binding and co-factor (Fe-II) binding. The 3D structures of flavanol synthase (FNS), 1-aminocyclopropane-1-carboxylic acid oxidases (ACOs), and gibberellin oxidases (GAOXs) were predicted, and docking interactions were studied with specific substrates. In addition, the expression levels of St2ODD genes were measured under abiotic stresses (salt, drought, and heat) to explore their roles with respect to each condition, which was further validated for salt and drought stress by qRT-PCR. Our results identified St2ODD genes in potatoes, showing changes in expression under drought and salt stress, and established a knowledge domain and theoretical basis for further improvement of potato and potato breeding.

Materials and Methods

Retrieval and identification of potential 2ODDs genes from potato

Firstly, the proteome of S. tuberosum was retrieved from Spud DB (http://spuddb.uga.edu) DM1-3 v6.1 (Pham et al., 2017). Reference sequences of 2ODDs of S. lycopersicum, S. chacoense, S. melongena, Manihot esculenta, Capsicum annuum, Arabidopsis thaliana, and Nicotiana tabacum were downloaded from the National Centre for Biotechnology Institute (NCBI) protein database, which are shown in Table S1. The reference genes were Blastp (e values of <0.001) searched against the potato proteome (Verma & Singh, 2021). Secondly, Hidden Markov Model (HMM) profiles of St2ODDs were fetched from the pfam database, and the retrieved sequences were searched (e value cut-off 1e-05) against HMM profiles. The candidate sequences obtained from both methods were considered putative 2ODDs genes and were further analysed for 2ODDs domains (2OG-FeII_Oxy, pfam03171 and DIOX_N, pfam14226) using various servers like SMART (http://smart.embl-heidelberg.de/) (Letunic, Khedkar & Bork, 2021), and NCBI Conserved Domain Search (https://www.ncbi.nlm.nih.gov/Structure/cdd/wrpsb.cgi) (Marchler-Bauer et al., 2015).

Gene structure, motif analysis and chromosomal mapping

The diversity of St2ODD genes was further analysed by studying the gene structure and conserved protein domains. To visualise the intron/exon structure, the online tool Gene Structure Display Server (GSDS) 2.0 (http://gsds.gao-lab.org/index.php) (Hu et al., 2015) was used. Conserved motifs were identified using the MEME online website (https://meme-suite.org/meme) with a maximum number of motifs, 10; optimum width of each motif, between 50 and 100 residues; other parameters were set to default values (Bailey et al., 2015). Chromosomal locations of 2ODDs were collected from Spud DB (http://spuddb.uga.edu) DM1-3 v6.1 (Pham et al., 2017), and were distributed across the 12 chromosomes of potato using TBtools (https://github.com/CJ-Chen/TBtools/releases) (Chen et al., 2020). Nomenclature was given based on the order of the chromosomal location of the genes.

Phylogenetic analysis and gene duplication

Multiple protein sequences of 205 identified genes were aligned using ClustalW in Molecular Evolutionary Genetics Analysis MEGA 7.0 (Kumar et al., 2016) (https://www.megasoftware.net/) and the phylogenetic tree was constructed with the neighbour-joining (NJ) method with 1,000 bootstrap replicates and complete deletion. The phylogenetic tree constructed was further analysed with the ITOL tool (https://itol.embl.de/upload.cgi) (Letunic & Bork, 2007). Gene duplication events were considered to have 80% or more identity with an e value 1e−10. The synonymous substitution (Ks) rates and nonsynonymous substitution (Ka) rates of duplicated 2ODD genes were calculated using Pal2nal http://www.bork.embl.de/pal2nal (Suyama, Torrents & Bork, 2006) and selection pressure was evaluated by calculating the Ka/Ks ratio (Shumayla et al., 2019).

Protein structure analysis and 3D modelling

The physio-chemical properties and subcellular localization of 2ODDs proteins were calculated using the ProtParam ExPasy server (https://web.expasy.org/protparam/) (Walker et al., 2005) and the ProtComp version 9.0 server (http://www.softberry.com) (Emanuelsson et al., 2000). SWISS-Model was used for 3D structure prediction of St2ODDs (Waterhouse et al., 2018) and visual representation of 3D structures of St2ODDs was done using UCSF Chimera (Pettersen et al., 2004). Ligand 3D models with PubChem CID 51 (2-Oxoglutaric acid), specific to FNS and GaOX, and PubChem CID 769 (Bicarbonate), specific to ACOs, were retrieved from the PubChem database in SDF format (https://pubchem.ncbi.nlm.nih.gov/) (Kim et al., 2016) and converted to PDB format using PyMOL (Schrödinger, LLC, New York, NY, USA).

Protein-ligand docking evaluation

The molecular docking of St2ODDs and their various ligands was performed using AutoDock 4. The 3D conformations of various ligands were retrieved from PubChem in SDF format: PubChem CID 51 (2-oxoglutaric acid) and PubChem CID 769 (bicarbonate). PyMOL Version 2.0 (Schrödinger, LLC, New York, NY, USA) was utilised for converting it to PDB format. The PDB files were then converted to PDBQT format using AutoDock 4 (Morris et al., 2009). Various parameters were assigned, including the addition of non-polar hydrogens and gasteiger charges. Grid boxes were generated with different dimensions in X,Y, and Z directions, as shown in Table S3. The proteins and ligands were docked with an energy range of 4 and exhaustiveness set to 8, and the best conformation was selected as having the lowest free energy (Anand et al., 2022). The protein-ligand hydrophobic and hydrogen bond interactions were represented with the BIOVIA Discovery Studio Visualizer (https://www.3ds.com/products-services/biovia/products/molecular-modeling-simulation/biovia-discovery-studio/).

Identification of abiotic stress responsive 2ODDs genes

Sequence Read Archive (SRA) data corresponding to project numbers SRP056128, SRP229183, and SRP237987 for drought, heat, and salt were downloaded from the NCBI SRA. Differentially expressed genes were identified using the Trinity-V 2.03 package. Transcript abundance was calculated in FPKM (fragments per kilobase of transcript per million mapped reads), and the heat map of expressed 2ODD genes under various stress conditions was visualised using TBtools (https://github.com/CJ-Chen/TBtools/releases) (Chen et al., 2020). Putative 2ODD genes responsive to stress were validated using RT-qPCR (Verma, Upadhyay & Singh, 2021).

Plant materials and treatments

Solanum tuberosum cv Kufri jyoti plantlets were received from the ICAR-Central Potato Research Institute (CPRI, Shimla) and maintained in a growth chamber in soil under a 16-hour (h) light/8-hour (h) dark photoperiod, 240 C, and 60% humidity. The seedlings were watered regularly. For drought stress, watering was withheld to mimic severe drought conditions for 3 days, followed by rewatering for 3 days. The leaves of drought-treated plants, after rewatering, and control plants were harvested immediately in liquid nitrogen and then stored at −80 °C. For salt stress, 4-week-old plants grown in soil were subjected to 500 mmol/L NaCl. The salt-treated plants were collected after 0, 24, 48, 72, and 96 h time courses. Collected leaf samples were stored immediately in liquid nitrogen, followed by −80 °C for further experimentation.

RNA isolation, cDNA preparation and qPCR analysis

Total RNA from the drought and salt stress samples, along with control leaf samples, was extracted from the samples (Ghawana et al., 2011). cDNA was synthesised using the Superscript III first-strand cDNA synthesis kit (Invitrogen USA) according to the manufacturer’s instructions. Elongation factor 1 alfa (ef1α) was the reference gene for expression normalization. The expression levels of St2ODD genes in potatoes subjected to drought and salt stress were measured using qRT-PCR. Primers were designed using Primer 3 software (http://primer3.ut.ee) (Untergasser et al., 2012) and the primer sequences are shown in Table S4. qRT-PCR was performed on stressed and control samples using the Bio-Rad CFX96 real-time PCR detection system. All the samples were taken in triplicate, and three technical replicates of each biological replicate were taken. PCR conditions were 95 °C for 5 min, followed by 38 cycles of 95 °C for 20 s, Tm for 20 s, and 72 °C for 20 s. Tm optimization of the stress-responsive St2ODDs was done using semi-quantitative PCR. The correlative expression data were calculated using the 2(−∆∆CT) method (Livak & Schmittgen, 2001).

Results

Identification of 2ODDs in potato

The 2ODDs belong to the 2OGD gene superfamily and are involved in various plant metabolic pathways, which have been explored in previous studies (You et al., 2019; Farrow et al., 2014; Hagel & Facchini, 2018). Here, to identify the 2ODDs in the potato whole proteome, BLASTP searched against the reference 2ODDs of related plants, and all the putative genes were retrieved and examined for the presence of domains associated with 2ODDs, i.e., 2OG-FeII_Oxy (PF03171) and DIOX_N (PF14226), using various bioinformatics tools like SMART and NCBI Conserved Domain Search. Eventually, a total of 205 St2ODDs were confirmed to have the 2ODD domains: 2OG-FeII_Oxy (PF03171) signature domain and DIOX_N (PF14226) domain. The physiochemical properties of these identified St2ODDs were studied, and the coding sequence (CDS) lengths, genomic sequence lengths, molecular weights (MWs), isoelectric points (PIs), and grand average of hydropathicity index (GRAVY) of these genes are shown in Table S2. The MWs of St2ODDs ranged from 22.3 kDa to 108.5 kDa. St2ODD30 had the smallest amino acid sequence of 198 aa and St2ODD54 had the largest amino acid sequence of 910 aa. The exonic numbers ranged from 2 to 14 exons, and the predicted PIs values ranged from 4.68 to 8.94, suggesting that St2ODDs consist of both basic and acidic proteins. The subcellular localization analysis suggested that St2ODDs in potatoes were expressed either in the cytoplasm or secreted extracellularly. St2ODDs consist of many plant secondary metabolites that are localised in the cytosol (Xu et al., 2008). This study predicted the same and gave insight into the basic aspects of St2ODDs.

Chromosomal distribution and gene duplication of St2ODD genes

The chromosomal location of 205 St2ODDs was identified using the Spud database, and a chromosomal distribution map of St2ODDs was constructed (Fig. 1). The chromosomal distribution showed that St2ODDs are widely distributed across the 12 chromosomes of potatoes. Chromosomes 2 and 9 have the maximum number of St2ODD genes (35/205) each. Chromosome 5 has the minimum number of St2ODDs (3/205). Our results suggest that most of the St2ODDs were distributed across the proximal ends of the chromosomes. For the expansion and generation of gene families, gene duplication plays a vital role, which needed to be studied for St2ODDs in S. tuberosum. Thus, we analysed gene duplication events for 2ODDs in S. tuberosum (Hofberger et al., 2015). Gene duplication events occur due to uneven crossing over chromosome duplication. The genes with 80% or more identity located on the same chromosome were considered duplication events. Tandem duplication is defined earlier as the duplication event within the 5 Mb region of a chromosome, while other duplication events are characterised as segmental duplication (Agarwal et al., 2016). Gene duplication events were analysed for St2ODDs genes, and the identified events were confirmed for tandem duplication and segmental duplication for their genome-wide expansion. On the basis of chromosomal location, gene clusters were observed on chromosomes 1, 2, 6, 7, 9, and 11 (Fig. 1). A total of 139 duplicated genes were found on the S. tuberosum genome, which showed that 67 percent of St2ODDs were originated by duplication events, i.e., tandem and segmental (Fig. 1). Further, fifty-six percent of duplication events were tandem duplications, and forty-four percent of duplication events were segmental duplications. This suggests that most of the expansion of St2ODDs in S. tuberosum is due to tandem duplication. The synonymous (Ks) and non-synonymous (Ka) values were calculated, and their ratios (Ka/Ks) depicted the nature of selection. The Ka/Ks ratio >1 depicts positive selection, Ka/Ks = 1 depicts neutral selection, and Ka/Ks <1 depicts purifying selection or negative selection. For St2ODDs, the Ka/Ks ratios were calculated and then evaluated to determine whether the selection pressure was the driving force for evolution or not (Liu et al., 2014). In this study, 134/139 duplicated St2ODDs experienced negative or purifying selection (Table 1), and among them, thirty-three pairs of duplication events of St2ODDs had Ka/Ks less than 0.3, suggesting their lower functional divergence during evolution. However, five pairs of duplicated St2ODDs experienced positive selection.

Figure 1 Schematic representation of the chromosomal distribution of St2ODD genes across the twelve chromosomes of potato and the black lines represent the duplicated gene pairs.

Table 1 Duplicated St2ODD gene pairs with nonsynonymous substitution (Ka) rates and synonymous substitution (Ks), Ka/Ks, selection type, and duplication type.

Dup Gene 1	Dup Gene 2	Ka	Ks	Ka/Ks	Selection	Duplication type	
St2ODD144	St2ODD4	0.06569628	0.04190762	1.56764532	Positive	TD	
St2ODD11	St2ODD7	0.00663721	0.01490115	0.44541598	Negative	SD	
St2ODD2	St2ODD6	0.09652673	0.1050776	0.9186233	Negative	SD	
St2ODD2	St2ODD5	0.00663721	0.01490115	0.44541598	Negative	SD	
St2ODD2	St2ODD8	0.0186345	0.02378455	0.78347065	Negative	SD	
St2ODD3	St2ODD4	0.06572274	0.04184643	1.57056977	Positive	TD	
St2ODD5	St2ODD8	0.00970602	0.01008984	0.9619597	Negative	TD	
St2ODD6	St2ODD7	0.08704825	0.08833728	0.98540787	Negative	TD	
St2ODD6	St2ODD8	0.00439755	0.01007854	0.43632812	Negative	TD	
St2ODD6	St2ODD9	0.08704825	0.08833728	0.98540787	Negative	TD	
St2ODD7	St2ODD8	0.00970602	0.01008984	0.9619597	Negative	TD	
St2ODD10	St2ODD15	0.05577109	0.19061954	0.29257806	Negative	SD	
St2ODD10	St2ODD29	0.0237322	0.08128247	0.29197198	Negative	SD	
St2ODD10	St2ODD30	0.04992497	0.1109401	0.45001735	Negative	SD	
St2ODD10	St2ODD82	0.0528703	0.18635514	0.28370723	Negative	SD	
St2ODD11	St2ODD12	0.06128759	0.15116171	0.40544385	Negative	TD	
St2ODD11	St2ODD188	0.07898557	0.23428441	0.33713543	Negative	SD	
St2ODD12	St2ODD188	0.036845	0.1478565	0.24919433	Negative	SD	
St2ODD15	St2ODD29	0.06293806	0.20941455	0.30054294	Negative	SD	
St2ODD15	St2ODD30	0.10259873	0.20577881	0.49858742	Negative	SD	
St2ODD15	St2ODD82	0.06273337	0.21128955	0.29690712	Negative	SD	
St2ODD18	St2ODD19	0.10511845	0.30308326	0.34683028	Negative	TD	
St2ODD29	St2ODD30	0.03556222	0.06253621	0.56866604	Negative	TD	
St2ODD29	St2ODD82	0.04824609	0.19460705	0.24791543	Negative	SD	
St2ODD30	St2ODD82	0.09859509	0.21095941	0.46736519	Negative	SD	
St2ODD31	St2ODD32	0.02620672	0.05593442	0.46852574	Negative	TD	
St2ODD33	St2ODD114	0.06716191	0.13402513	0.50111431	Negative	SD	
St2ODD42	St2ODD45	0.06402212	0.19993341	0.32021722	Negative	TD	
St2ODD44	St2ODD47	0.07057711	0.29750572	0.23722942	Negative	TD	
St2ODD44	St2ODD48	0.09400143	0.19247944	0.48837128	Negative	TD	
St2ODD46	St2ODD53	0.16970163	0.40972685	0.41418235	Negative	TD	
St2ODD47	St2ODD48	0.14488402	0.38971926	0.37176511	Negative	TD	
St2ODD50	St2ODD51	0.12069077	0.34853361	0.34628157	Negative	TD	
St2ODD51	St2ODD53	0.17161054	0.46632126	0.36800925	Negative	TD	
St2ODD53	St2ODD54	0.35046972	0.81523819	0.42989856	Negative	TD	
St2ODD77	St2ODD80	0.08738575	0.2360608	0.37018323	Negative	TD	
St2ODD84	St2ODD85	0.04315687	0.186145	0.23184547	Negative	TD	
St2ODD84	St2ODD86	0.04954394	0.20793712	0.23826406	Negative	TD	
St2ODD84	St2ODD87	0.04834956	0.19757062	0.2447204	Negative	TD	
St2ODD84	St2ODD190	0.04339296	0.12568765	0.34524439	Negative	SD	
St2ODD84	St2ODD193	0.05668885	0.14432865	0.39277613	Negative	SD	
St2ODD84	St2ODD194	0.07353332	0.256353	0.28684399	Negative	SD	
St2ODD84	St2ODD195	0.0561702	0.15104611	0.37187453	Negative	SD	
St2ODD84	St2ODD196	0.06644118	0.26447813	0.25121618	Negative	SD	
St2ODD84	St2ODD197	0.07020124	0.25027297	0.28049867	Negative	SD	
St2ODD84	St2ODD198	0.06255558	0.270784	0.23101655	Negative	SD	
St2ODD85	St2ODD87	0.01434164	0.01789127	0.8015998	Negative	TD	
St2ODD85	St2ODD86	0.0223309	0.01945004	1.14811592	Positive	TD	
St2ODD85	St2ODD190	0.05515156	0.15103153	0.36516583	Negative	SD	
St2ODD85	St2ODD88	0.02882001	0.04695144	0.61382592	Negative	TD	
St2ODD85	St2ODD195	0.07291675	0.19592791	0.3721611	Negative	SD	
St2ODD85	St2ODD193	0.06928318	0.19005848	0.3645361	Negative	SD	
St2ODD85	St2ODD198	0.0776672	0.26823273	0.28955153	Negative	SD	
St2ODD85	St2ODD194	0.08839285	0.27488677	0.32156095	Negative	SD	
St2ODD85	St2ODD196	0.08118562	0.27035025	0.30029793	Negative	SD	
St2ODD85	St2ODD197	0.09097925	0.26529128	0.34294097	Negative	SD	
St2ODD86	St2ODD87	0.02496672	0.01955304	1.27687138	Positive	TD	
St2ODD86	St2ODD88	0.04117256	0.04574588	0.90002773	Negative	TD	
St2ODD86	St2ODD190	0.06137565	0.16833057	0.36461381	Negative	SD	
St2ODD86	St2ODD193	0.07485809	0.17729158	0.42223152	Negative	SD	
St2ODD86	St2ODD194	0.09216987	0.265857	0.34668965	Negative	SD	
St2ODD86	St2ODD195	0.07931784	0.21289575	0.37256657	Negative	SD	
St2ODD86	St2ODD196	0.0857194	0.27485218	0.31187456	Negative	SD	
St2ODD86	St2ODD197	0.09645194	0.3121203	0.30902169	Negative	SD	
St2ODD86	St2ODD198	0.0833017	0.26559101	0.31364654	Negative	SD	
St2ODD87	St2ODD88	0.0361952	0.04295253	0.84267917	Negative	TD	
St2ODD87	St2ODD190	0.06040845	0.16179674	0.37336008	Negative	SD	
St2ODD87	St2ODD193	0.07461881	0.2016231	0.37009057	Negative	SD	
St2ODD87	St2ODD194	0.09053363	0.27815609	0.32547781	Negative	SD	
St2ODD87	St2ODD195	0.07806541	0.20848701	0.37443779	Negative	SD	
St2ODD87	St2ODD196	0.08464087	0.27357136	0.30939228	Negative	SD	
St2ODD87	St2ODD197	0.09643984	0.27850651	0.34627499	Negative	SD	
St2ODD87	St2ODD198	0.08239731	0.27192378	0.3030162	Negative	SD	
St2ODD88	St2ODD190	0.06748491	0.15156264	0.44526081	Negative	SD	
St2ODD88	St2ODD193	0.0784197	0.20164162	0.38890631	Negative	SD	
St2ODD88	St2ODD194	0.10073583	0.23401431	0.43046866	Negative	SD	
St2ODD88	St2ODD195	0.08762846	0.18971131	0.46190424	Negative	SD	
St2ODD88	St2ODD196	0.09043122	0.25011537	0.36155802	Negative	SD	
St2ODD88	St2ODD197	0.11408173	0.24645762	0.4628858	Negative	SD	
St2ODD88	St2ODD198	0.09807243	0.22515042	0.43558626	Negative	SD	
St2ODD97	St2ODD101	0.15082731	0.28041998	0.5378622	Negative	TD	
St2ODD97	St2ODD103	0.1491623	0.29948343	0.49806528	Negative	TD	
St2ODD97	St2ODD105	0.14054039	0.27248961	0.51576422	Negative	TD	
St2ODD98	St2ODD144	0.02450309	0.07493749	0.32698035	Negative	SD	
St2ODD98	St2ODD175	0.19572533	0.36390979	0.53784025	Negative	SD	
St2ODD101	St2ODD103	0.150478	0.30131704	0.4994009	Negative	TD	
St2ODD101	St2ODD105	0.04306739	0.0408264	1.05489048	Positive	TD	
St2ODD103	St2ODD105	0.15353419	0.31519188	0.48711341	Negative	TD	
St2ODD108	St2ODD109	0.06700429	0.10432615	0.64225783	Negative	TD	
St2ODD113	St2ODD189	0.66649649	1.77888311	0.37467133	Negative	SD	
St2ODD118	St2ODD120	0.0261915	0.2951278	0.08874629	Negative	TD	
St2ODD124	St2ODD126	0.10690827	0.4964094	0.2153631	Negative	TD	
St2ODD126	St2ODD128	0.05344949	0.20377605	0.26229523	Negative	TD	
St2ODD129	St2ODD130	0.05101614	0.11788919	0.4327466	Negative	TD	
St2ODD135	St2ODD136	0.21553725	0.34973463	0.61628799	Negative	TD	
St2ODD135	St2ODD141	0.10181782	0.19313922	0.52717322	Negative	TD	
St2ODD135	St2ODD142	0.09506626	0.18714606	0.50797894	Negative	TD	
St2ODD136	St2ODD141	0.11602076	0.28011867	0.4141843	Negative	TD	
St2ODD136	St2ODD142	0.1067933	0.28093034	0.38014157	Negative	TD	
St2ODD141	St2ODD142	0.01818822	0.02946714	0.61723748	Negative	TD	
St2ODD144	St2ODD175	0.07633702	0.1946334	0.39220926	Negative	SD	
St2ODD145	St2ODD146	0.00756468	0.03783308	0.19994879	Negative	TD	
St2ODD151	St2ODD155	0.08795754	0.26721516	0.32916373	Negative	TD	
St2ODD158	St2ODD162	0.14084357	0.25515971	0.55198202	Negative	TD	
St2ODD158	St2ODD163	0.10637851	0.19789956	0.53753791	Negative	TD	
St2ODD158	St2ODD164	0.12623217	0.15711323	0.80344711	Negative	TD	
St2ODD158	St2ODD166	0.09902427	0.20454306	0.4841243	Negative	TD	
St2ODD161	St2ODD162	0.1484228	0.35333191	0.42006623	Negative	TD	
St2ODD161	St2ODD163	0.09515891	0.29623739	0.32122519	Negative	TD	
St2ODD161	St2ODD165	0.15729244	0.33400836	0.47092367	Negative	TD	
St2ODD161	St2ODD166	0.08157109	0.3142541	0.25957048	Negative	TD	
St2ODD162	St2ODD163	0.09001811	0.17228699	0.52248933	Negative	TD	
St2ODD162	St2ODD166	0.08374729	0.15777214	0.53081162	Negative	TD	
St2ODD162	St2ODD165	0.10199099	0.18093047	0.56370268	Negative	TD	
St2ODD163	St2ODD166	0.04675867	0.12542874	0.37279076	Negative	TD	
St2ODD163	St2ODD164	0.0975672	0.18354877	0.53156007	Negative	TD	
St2ODD164	St2ODD166	0.09209668	0.20413856	0.4511479	Negative	TD	
St2ODD179	St2ODD180	0.0794928	0.16823277	0.47251674	Negative	TD	
St2ODD190	St2ODD193	0.04521834	0.15013735	0.30117979	Negative	SD	
St2ODD190	St2ODD194	0.06695894	0.24483643	0.27348438	Negative	SD	
St2ODD190	St2ODD195	0.04405385	0.16497941	0.26702635	Negative	SD	
St2ODD190	St2ODD196	0.05864168	0.22252378	0.26352996	Negative	SD	
St2ODD190	St2ODD197	0.06108833	0.24501942	0.24932036	Negative	SD	
St2ODD190	St2ODD198	0.05607785	0.22850289	0.24541419	Negative	SD	
St2ODD193	St2ODD194	0.06368396	0.20051302	0.31760512	Negative	TD	
St2ODD193	St2ODD195	0.02291405	0.09777482	0.23435536	Negative	TD	
St2ODD193	St2ODD196	0.05668303	0.22011799	0.25751201	Negative	TD	
St2ODD193	St2ODD197	0.06233667	0.23956419	0.26020865	Negative	TD	
St2ODD193	St2ODD198	0.05412739	0.19669002	0.27519131	Negative	TD	
St2ODD194	St2ODD195	0.06576495	0.24323853	0.27037227	Negative	TD	
St2ODD194	St2ODD196	0.03951081	0.08266321	0.47797329	Negative	TD	
St2ODD194	St2ODD197	0.07011406	0.18379532	0.38147899	Negative	TD	
St2ODD194	St2ODD198	0.02650612	0.03671861	0.72187153	Negative	TD	
St2ODD195	St2ODD196	0.05616442	0.23301649	0.24103195	Negative	TD	
St2ODD195	St2ODD197	0.05797008	0.24041471	0.24112535	Negative	TD	
St2ODD195	St2ODD198	0.05105475	0.23904667	0.21357651	Negative	TD	
St2ODD196	St2ODD197	0.05664229	0.20893578	0.27109905	Negative	TD	
St2ODD196	St2ODD198	0.02408395	0.0849296	0.28357542	Negative	TD	
St2ODD197	St2ODD198	0.05920475	0.18579026	0.31866447	Negative	TD	

Phylogenetic analysis of 2ODDs in potato

To understand the phylogenetic relationship between all the identified St2ODDs, a phylogenetic tree was constructed. Based on the phylogenetic tree, St2ODDs could be divided into eight groups (1–8) based on their functional aspects (Fig. 2), which were based on gene ontology studies and homology with characterised A. thaliana proteins. Amongst them, the largest group contained 2-oxoglutarate dioxygenases (2OGs). While 38 genes had gibberellin oxidase (GAOX) functions, Flavanone synthase (FNS) function was observed in 22 genes, and 10 genes have 1-aminocyclopropane-1-carboxylic acid oxidases (ACOs) function. Other functional groups were observed, including downy mildew resistance 6 (DMR6), senescence-related genes (SRGs), dioxygenase for auxin oxidation, and jasmonate-induced oxygenase genes.

Figure 2 Phylogenetic analysis of St2ODDs in potato.

The St2ODD amino acid sequences were aligned using ClustalW, the phylogenetic tree was constructed using neighbour-joining (NJ) method with 1,000 bootstrap and complete deletion in MEGA 7.

Gene structure and motif analysis of St2ODDs

St2ODDs have a diversified gene structure, as the exonic number of the identified St2ODDs varied from two to 14 in number (Fig. 3A). St2ODD54 contained the maximum number of exons (14), according to Fig. 3. Next, the protein sequences of all St2ODDs were examined for the presence of different motifs using Multiple Expectation Maximisation for Motif Elicitation (MEME). We identified ten different conserved motifs, named motif 1 to motif 10 (Fig. 4), dispersed throughout the protein sequences. Four motifs (motifs 1, 2, 3, and 4) were distributed throughout the St2ODDs, and motifs 5–10 were present specifically in different clusters. Similar clusters showed similar distributions of motifs based on their functions. The conserved motifs and their sequences are shown in Table S5.

Figure 3 Genestructures of St2ODDs.

Exons, introns, and upstream/downstream are represented by blue, black lines, and orange, respectively.

Figure 4 Motif composition in St2ODDs.

St2ODDs conserved motifs were represented in different colored boxes: motif 1 (red), motif 2 (cyan), motif 3 (light green), motif 4 (purple), motif 5 (mustard), motif 6 (dark green), motif 7 (navy), motif 8 (pink), motif 9 (orange), and motif 10 (yellow).

Further identified motifs were analysed, and motif 6 contained active site amino acid residues asparagine (N), tyrosine (Y), histidine (H) and aspartic acid (D). Motif 1 contained the active site amino acid residues histidine (H), arginine (R) and serine (S) which are conserved throughout the eight groups, showing the importance of motif 6 and motif 1 in the St2ODD enzyme family for functioning. The amino acids N, Y, R, and S are responsible for interacting with 2 OG, and the amino acids H, H, and D are responsible for interacting with Fe (II), which is a cofactor (Takehara et al., 2020). The conserved active amino acid residues suggest their potential role in interactions with their substrate and the functioning of these identified St2ODDs.

Protein-ligand interactions of St2ODDs

The 3D structures of representative members of FNS, GaOx, and ACOs in each group based on their function were modelled using SWISS-MODEL on the basis of homology. All the selected groups have vital roles in flavonoid biosynthesis, gibberellin biosynthesis, and ethylene biosynthesis, which needed to be studied for further understanding the substrate binding and functioning of these proteins, as shown in Fig. 5. The active amino acid residues for 2OG binding and iron co-factor binding with FNS were previously described (Sun et al., 2015). Similarly, the active amino acid residues responsible for 2OG and iron co-factor binding with GaOx were previously described (Takehara et al., 2020). For ACOs, the active residues were previously described and used as references for finding the active amino acid residues in St2ODDs (Zhang et al., 2004). The multiple sequence alignment of the three groups having FNS, GaOx, and ACO function was made and visualised, in which the active amino acids N, Y, R, and S responsible for interacting with 2 OG were conserved for binding with FNS and GaOx. For ACO, the binding sites for bicarbonate were also conserved, i.e., R and R. However, the co-factor binding, i.e., Fe-II, and the amino acids (H, H, and D) were conserved in the three groups, as shown in Fig. 6. These conserved residues suggest the functional importance of these residues for possible interactions with substrates and co-factors. In this study, a few St2ODDs have altered active amino acid residues, inferring their possible functional divergence. To further validate the involvement of these active amino acid residues, docking was performed. Hydrogen and hydrophobic interactions were studied between St2ODD29, St2ODD124 of GaOXs, and 2-Oxoglutaric acid (CID 51). St2ODD85, St2ODD87 of FNS, and 2-Oxoglutaric acid (CID 51). St2ODD118, St2ODD120 of ACOs and bicarbonate (CID 769) (Figs. 7A–7B) The docking results of St2ODD29 and 2-Oxoglutaric acid showed hydrogen bond interactions with R292 and hydrophobic interactions with S294, Q234, and E94. The docking results of St2ODD124 and 2-Oxoglutaric acid showed hydrogen bond interactions with R263, S265 and hydrophobic interactions with L205, S207. Interactions of St2ODD118 and bicarbonate showed hydrogen bonds with R244, Q239, and hydrophobic interactions with A238, Y162. Interactions of St2ODD120 and bicarbonate showed hydrogen bonds with R235 and hydrophobic interactions with R175, H234, and H177 (Figs. 7C–7D). The docking results of St2ODD85 and 2-Oxoglutaric acid showed hydrogen bond interactions with S317 and hydrophobic interactions with R315, Y232. The docking results of St2ODD87 and 2-Oxoglutaric acid showed hydrogen bond interactions with S317, L264, and I316 and hydrophobic interactions with R315 (Figs. 7E–7F). These results validated the role of the conserved amino acids in stabilising and interacting with the substrates responsible for leading to the plant pathways.

Figure 5 A schematic representation of the 3D structure of representative members of (A and B) GaOx (St2ODD29 and St2ODD124). 2-OG binding sites Y, R, and S are represented in red color, while Fe(II) binding sites H, D, and H are represented in blue color. (C and D) ACC (St2ODD118 and St2ODD120). Bicarbonate binding sites R, and R are represented in pink color, while Fe(II) binding sites H, D, and H are represented in blue color. (E and F) FNS (St2ODD85 and St2ODD87).

2-OG binding sites N, Y, R, and S are represented in red color, while Fe(II) binding sites H, H, and D are represented in blue color.

Figure 6 Multiple alignment of (A) GaOx function containing St2ODDs. Active amino acids tyrosine (Y), arginine (R), and serine (S) responsible for interacting with 2 OG are represented with blue triangles. The active residues histidine (H), histidine (H) and aspartate (D) responsible for interacting with Fe (II) are represented with red triangles. (B) FNS function containing St2ODDs. Active amino acids asparagine (N), tyrosine (Y), arginine (R), and serine (S) responsible for interacting with 2 OG are represented with blue triangles. The active residues histidine (H), histidine (H) and aspartate (D) responsible for interacting with Fe (II) are represented with red triangles. (C) ACC function containing St2ODDs.

V Active amino acids arginine (R), and arginine (R) responsible for interacting with bicarbonate are represented with yellow triangles. The active residues histidine (H), histidine (H) and aspartate (D) responsible for interacting with Fe (II) are represented with red triangles.

Figure 7 Hydrogen and hydrophobic interaction profile of (A and B) St2ODD29 and St2ODD124 with 2-oxoglutaric acid (2OG) respectively. (C and D) St2ODD118 and St2ODD120 with bicarbonate respectively. (E and F) St2ODD85 and St2ODD87 with 2OG. The analysis was done using Discovery studio and represented the residues involved in interaction.

Expression pattern of St2ODDs under drought, salt, and heat stresses

Abiotic stresses play a crucial role in the growth and yield of the potato, and to explore the potential roles of St2ODDs against these abiotic stresses, expression levels of St2ODDs under drought, salt, and heat treatments were measured using available RNA-seq data. On the basis of log 2 fragments per kilobase of transcript per million fragments (FPKM) values, expression patterns were identified (Fig. 8). Under salt stress, 71 genes were differentially expressed: 39% (15/38) of gibberellin oxidases (GAOXs), which play a part in GA biosynthesis and the catabolism pathway, are expressed in salt stress (Hu et al., 2021) and 30% (3/10) of 1-aminocyclopropane-1-carboxylic acid oxidases (ACOs), which play a part in ethylene biosynthesis, are expressed in salt stress (Chang et al., 2019) (Fig. 7). Under heat stress, seven genes were upregulated, among them St2ODD113, which belongs to GAOXs, and fifteen genes were downregulated. St2ODD125 and St2ODD119 encode GAOXs and ACOs, respectively (Fig. 7). Under drought stress, twenty-nine genes were differentially expressed. St2ODD130, St2ODD54, and St2ODD25 were upregulated. Thirteen genes were downregulated under drought stress. St2ODD73, St2ODD10, St2ODD34, St2ODD99, St2ODD127, St2ODD17, St2ODD125, and St2ODD30 belong to the GaOx group. St2ODD115 and St2ODD56 belong to the ACOs group. St2ODD195, St2ODD189, and St2ODD193 belong to the FNS group. The expression patterns of the St2ODD gene family suggest their role in various stress conditions, and some of them are downregulating or upregulating in all three stresses or in groups suggesting their role in accordance with abiotic stress.

Figure 8 Differential expression profiling of St2ODDs under abiotic stressors: (A) drought stress, (B) salt stress and (C) heat stress.

Effect of St2ODDs expression under drought and salt stress

To evaluate the role of St2ODDs under drought stress and salt stress, the gene expression levels of nine St2ODDs were measured. The gene expression levels of St2ODD54, St2ODD25, St2ODD130, St2ODD22, and St2ODD112 were measured under drought treatment after 3 days and in rewatering after 3 days by qRT-PCR. For salt treatment, the gene expression levels of St2ODD138, St2ODD76, St2ODD91, and St2ODD34 were measured after 24, 48, 72, and 96 h. The genes were selected based on the FPKM values; the genes with higher values were selected and validated via qRT-PCR.

The expression levels of St2ODDs under drought stress were analysed, and St2ODD130, St2ODD54, and St2ODD25 showed increased relative expression under drought stress with 2.26 fold change (FC), 1.71 fold change (FC), and 2.5 fold change (FC), respectively (Fig. 9A). However, the relative expression of St2ODD22 and St2ODD112 was downregulated in the drought condition with 0.47 FC and 0.45 FC, respectively, but showed a significant change in FC after rewatering.

Figure 9 Real-time quantitative reverse transcription-polymerase chain reaction (qRT-PCR) analyses of St2ODDs in plants. (A) Under drought stress for the drought (DRT), rewatering (RWT) conditions and are shown on the x-axis and the fold change on the y-axis. (B) Under salt stress at different time points 24, 48, 72, 96 h and are shown on the x-axis and the fold change on the y-axis.

The data were analysed by three biological repeats, and represented with mean ± SD where ns means p > 0.05, ** means p < 0.01, and *** means p < 0.001. Raw data for qRT-PCR for drought and salt stress are given in Table S5.

Under salt stress, the expression levels of the St2ODDs were measured in 1-month-old seedlings at different time points, i.e., 24, 48, 72, and 96 h. Four genes showed increased expression levels, i.e., St2ODD138, St2ODD76, St2ODD91, and St2ODD34, throughout the salt stress, which is in accordance with the expression patterns of the available RNASeq data (Fig. 9B). St2ODD138 was upregulated with FC of 1.77 after 24 h, 3.04 FC after 48 h, 4.1 FC after 72 h, and 5.71 FC after 96 h. St2ODD76 was upregulated with FC of 1.42 after 24 h, 3.44 FC after 48 h, 5.58 FC after 72 h, and 8.66 FC after 96 h. St2ODD91 was upregulated consistently throughout, with FC of 3.61 after 24 h, 5.18 FC after 48 h, 7.6 FC after 72 h, and 16.7 FC after 96 h (Fig. 8B). St2ODD34 was upregulated with FC of 1.35 after 24 h, 1.99 FC after 48 h, 3.3 FC after 72 h, and 3.71 FC after 96 h.

The elevated relative expression levels suggest the potential involvement of St2ODDs with respect to abiotic stresses, which pose huge agricultural losses globally to the potato.

Discussion

The potato is the world’s third most important crop and is of immense importance to humans. Environmental stresses influence its growth, tuber size, and tuber number (Schafleitner et al., 2007; Eiasu, Soundy & Hammes, 2007). The potato yield is affected by various abiotic factors, including salinity, drought, and heat (Deblonde & Ledent, 2001; Levy & Veilleux, 2007). The 2ODDs are members of the 2OGDs superfamily, the second-largest plant family. They play a crucial role in a variety of biological processes involved in plant metabolism, such as gibberellin biosynthesis, ethylene biosynthesis, and auxin catabolism, among others, are involved in secondary metabolism, and are the subject of extensive research (Farrow et al., 2014). However, 2ODDs are also elevated in response to diverse abiotic stresses, such as cold, salt, and drought (Mahajan, Sudesh & Yadav, 2014; Meng et al., 2015; Wang et al., 2020). The genome-wide analysis is an essential method for identifying the biological functions of the St2ODD gene family in potatoes. The 2ODDs have been previously characterised in S. lycopersicum and C. sativa; however, the St2ODDs have not been previously characterised in S. tuberosum (Potato) (Wei et al., 2021; Zhu et al., 2022).

The approach for identifying St2ODDs using BlastP searches the known proteins of related families against the potato proteome. HMM profiles of St2ODDs from the Pfam database were searched against HMM profiles and followed as reported earlier (Shumayla et al., 2019; Verma & Singh, 2021). Our study promotes the evolutionary, functional, structural, and interaction aspects of St2ODD family genes with different substrates in potatoes. Here, we identified a total of 205 2ODDs in potato, which is comparable to the 131 Sl2ODDs identified by Wei et al. (2021) in tomato, which have various functions like ACOs, GAOxs, F3H, etc. The identified St2ODDs were evenly distributed across the 12 chromosomes of potato, and the gene duplication events, i.e., tandem and segmental duplication, revealed that 56% of duplication events are due to tandem duplications. Moreover, sixty-seven percent of St2ODDs family expansion is due to duplication events. The non-synonymous (Ka) and synonymous (Ks) values and their ratios were calculated to determine the nature of selection for the expansion of St2ODDs in potatoes. The results infer that most of the St2ODDs were expanded as a result of negative selection or purifying selection and had lower functional divergence.

The identified St2ODDs were analysed for evolutionary relationships and, based on their functions, classified into eight groups: ACOs, involved in ethylene biosynthesis (Chang et al., 2019), GAOxs, involved in gibberellin biosynthesis (Huang et al., 2015), FNS, involved in the flavonoid biosynthesis pathway (Tohge, Perez De Souza & Fernie, 2017). The classification is consistent with the previous studies (Wei et al., 2021). All the identified St2ODDs have conserved 2OG-FeII_Oxy (PF03171) signature domains and also have the conserved DIOX_N (PF14226) domain, which is a characteristic of the 2ODD superfamily (Kawai, Ono & Mizutani, 2014). Motif analysis showed that similar clusters have similar motif structures, and Motifs 1, 2, 3, and 6 were present throughout the St2ODDs, inferring that they have important roles in the activity of the St2ODDs. Motif 6 contained active site amino acid residues Y, H, and D and motif 1 contained active site amino acid residues H, R, and S, inferring their roles in binding with a specific substrate and cofactor, i.e., Fe(II). A few motifs were specific to certain genes, like motif 8 and 10, which were specific to St2ODD203, St2ODD187, St2ODD93, etc. The roles of these specific motifs further need to be studied for their precise relationship, and they may have functional specificity in different classes.

The protein sequences of the identified St2ODDs were aligned, and the active amino sites were predicted. The alignment showed that St2ODDs possessed (HxDxnH and YxnRxS) motifs specific to Fe(II) and 2ODD binding (Takehara et al., 2020). Similarly, the sequence alignment of FLS1 (flavonol synthase), belonging to the 2ODD superfamily in various plants, including T. aestivum, A. thaliana, Dendrobium officinale etc. showed similar active amino acid residues (Yu et al., 2021). The active amino acid residues were also conserved in ANS (anthocyanidin synthase), belonging to the 2ODD (Xu et al., 2008), inferring their functional importance for the functioning of these enzymes. However, few St2ODDs didn’t have the conserved active amino acid residues, suggesting plausible alterations in their functionality during evolution. The protein 3D structures of the St2ODDs representative members of GaOx, FNS, and ACO were made and visualised for active residues Y, R, and S and RxR (Fig. 5) responsible for the binding with 2OG and bicarbonate, respectively. All the represented members had residues in close vicinity in the protein 3D structures, as shown in earlier studies (Takehara et al., 2020) which confirms the presence of binding sites for the substrate. After docking with specific substrates, results showed potential hydrogen and hydrophobic interactions between St2ODD29, St2ODD124 of GaOXs and 2-Oxoglutaric acid (CID 51). St2ODD85, St2ODD87 of FNS and 2-Oxoglutaric acid (CID 51). St2ODD118, St2ODD120 of ACOs and bicarbonate (CID 769). Various hydrophobic interactions were involved, which may facilitate the stabilisation of the complex. Our results suggest the presence of conserved residues in St2ODDs and their potential interactions with their respective substrates. The physiochemical properties of these identified St2ODDs were analysed, and various properties like amino acid residues, molecular weight, and exonic number, i.e., 2 to 14 in number, were in compliance with the previous studies (Wei et al., 2021) which infers the similarity in physiochemical properties of the 2ODDs across plants.

Abiotic stresses play an important role in the growth and yield of the plant. Potatoes are influenced by various abiotic stresses, including heat, salt, and drought stress. Heat stress depends on the degree and duration of heat, which triggers the production of reactive oxygen species (ROS) and oxidative stress, which alters the metabolism, growth, and productivity of plants (Hasanuzzaman et al., 2013). Salt stress is responsible for hindering plant growth, photosynthesis, and germination as well (Wang et al., 2021). Drought stress also hinders plant growth and affects photosynthesis due to stomatal closure. Plant growth hormones like auxins, gibberellins, etc. also modulate responses to drought (Farooq et al., 2012) and 2ODDs are responsible for the biosynthesis of these hormones, ultimately affecting plant responses to these stresses.

In this study, the expression patterns of St2ODDs were observed under heat, salt, and drought stress. The results showed an upregulated response of 34 genes when administered with salt stress. Out of 71 differentially expressed genes, 15 belong to the GAOXs and three belong to the ACOs, which are involved in the biosynthesis of gibberellins and ethylene subsequently. Various 2ODDs have been expressed in plant systems to check their role with administered abiotic stresses. Overexpressed F3H from Camellia sinensis in tobacco showed increased salt tolerance (Mahajan, Sudesh & Yadav, 2014) similarly, FNS1 (T. aestivum), belonging to the 2ODD superfamily, was overexpressed in A. thaliana and showed an increased root length under salt stress.

Under heat stress, seven genes were upregulated after 6 h but downregulated after 3 days, while two genes were upregulated throughout. Fifteen genes were downregulated under heat stress. Previous studies have reported the role of 2ODDs in heat stress protection. Overexpression of the F3HL protein of S. lycopersicum in tobacco reported increased expression levels of SlF3HL under chilling stress and faster seed germination growth levels when compared to control (Meng et al., 2015). Likewise, the identified genes associated with heat stress may be associated with heat stress protection and can be studied.

Under drought stress, twenty-nine genes were differentially expressed: St2ODD130, St2ODD54, and St2ODD25 were upregulated. Thirteen genes were downregulated under drought stress. Overexpression studies in A. thaliana showed FNS1 of P. nutans and LDOX2 of Reaumuria trigyna conferred increased resistance to drought by improving antioxidant capacity and increased tolerance to drought and UV-B subsequently in A. thaliana (Wang et al., 2020; Li et al., 2021). The expression patterns may suggest the role of St2ODDs in the elevated abiotic stresses.

The expression levels of St2ODDs were checked using qRT-PCR under drought stress after 3 days and in rewatering after 3 days, in which St2ODD130, St2ODD54, and St2ODD25 showed increased relative expression under drought stress and normal expression after rewatering. The expression levels of St2ODDs were also checked using qRT-PCR under salt stress after 24, 48, 72, and 96 h. Four St2ODDs showed increased expression levels, which is in compliance with the expression patterns of RNAseq data. All these results infer the potential roles of the identified St2ODDs in potato under different abiotic stresses, which further needs to be studied using overexpression systems and could be used for genome editing studies for improving desired traits like tolerance to abiotic stresses in potato. Overall, our study forms a knowledge base and provides functional insights into the 2ODD gene family in potatoes, which can be further explored to identify new aspects of 2ODDs. The roles of the identified St2ODDs could further be validated by in vitro studies. Overexpression and gene knock-down studies using designer nucleases like CRISPR/Cas could be used to validate the potential roles of St2ODDs in different abiotic stresses (Alok et al., 2021).

Conclusion

In conclusion, our findings provided understanding of the St2ODDs, which had not been previously investigated. Chromosomal locations, gene architectures, motif analysis, gene duplications, and evolutionary links were among the many elements of St2ODDs that were examined; these findings were consistent with earlier research. The presence of conserved active sites and their possible interactions with substrates were investigated and validated in relation to the structural features of St2ODDs. The docking and structural domains of identified St2ODDs could be useful for further research. Some candidate genes having roles in various abiotic stresses like heat, salt, and drought have been identified in accordance with the expression pattern, which was further validated for drought stress and salt stress by qRT-PCR. St2ODD130, St2ODD25, and St2ODD54 were found to be upregulated under drought stress, and St2ODD76, St2ODD91, St2ODD138, and St2ODD34 showed significant FC under salt stress. The identified genes having roles in association with abiotic stresses, i.e., drought stress and salt stress, can be further explored. Overexpression studies using CRISPR/Cas could further validate the role of St2ODDs.

Supplemental Information

Supplemental Information 1 Reference sequence IDs retrieved from database.

Click here for additional data file.

Supplemental Information 2 Predicted sequence features of St2ODDs.

Click here for additional data file.

Supplemental Information 3 Grid box dimension co-ordinates (X, Y, and Z) and binding affinity in Kcal/mol of the ten groups (1-10).

Click here for additional data file.

Supplemental Information 4 Primers used for qRT-PCR.

Click here for additional data file.

Supplemental Information 5 Sequences of conserved motifs (1-10) in St2ODDs.

Click here for additional data file.

Supplemental Information 6 Raw Data for qRT-PCR result for drought stress and salt stress.

Click here for additional data file.

The authors are thankful to the Department of Biotechnology at Panjab University for providing the necessary resources and infrastructure for conducting the required experiments. The authors are thankful to the Potato Genome Sequence Consortium (PGSC) for data availability.

Additional Information and Declarations

Competing Interests

Author Contributions

Data Availability

Kashmir Singh is an Academic Editor for PeerJ.

Hanny Chauhan performed the experiments, analyzed the data, prepared figures and/or tables, authored or reviewed drafts of the article, and approved the final draft.

Aiana performed the experiments, prepared figures and/or tables, and approved the final draft.

Kashmir Singh conceived and designed the experiments, authored or reviewed drafts of the article, and approved the final draft.

The following information was supplied regarding data availability:

The raw data is available in the Supplemental Files.

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
