# Peer review of "Genome-wide identification of 2-oxoglutarate and Fe (II)-dependent dioxygenase family genes and their expression profiling under drought and salt stress in potato"

_PeerJ, doi:10.7717/peerj.16449_

## Round 0.1 · original submission · Major Revisions

All of the reviewers found the work interesting however they expressed some concerns that the authors should resolve before it is considered for publication. Authors must, in particular, improve the manuscript writing and redo data analysis in various sections. I would advise authors to use appropriate and recent references as indicated by the reviewers to validate their analyses and findings.

Reviewer 1 ·

Basic reporting

It is an interesting and a comprehensive study on the molecular patterns of the dioxygenase family in Potato.
1. They have done computational and experimental studies but the title does not cover them. A broader title is suggested
2. The research question is not clear
3. Line 63, what do you mean by ‘it”?
4. Line 105, the link is not working
5. Line 169, 261, the references’ years are missed
6. Long sentences have to restructured
7. Where did authors get all the sequences? it has to mentioned in the materials and methods

Experimental design

In General, the experimental design and the data analysis are acceptable. However, correlations between all abiotic stresses in 2,6 and 2,7, and 3.6 and 3.7 sections are of value. This will help to understand interaction effects of abiotic stresses on gene expressions of 2OG family.

Validity of the findings

Findings are valid and valuable but the main findings have to highlighted and integrated, which helps to effectively conclude the novelty of this study

Additional comments

1. Discussion part is generally weak
2. Coherence is missed in discussion
3. Recent references are required as supporting evidence
4. Conclusion is weak, some recent evidences are also required to support the main finding

Reviewer 2 ·

Basic reporting

The language of manuscript should be reworked by professional.

Experimental design

The description of Plant materials and Experimental treatment is not clear. How to performed drought and salt stress. The biological duplication of the sample is not stated. The reference genes in qRT-PCR experiments are generally two genes.

Validity of the findings

no comment

Additional comments

The authors have identified the gene members of 2-Oxoglutatrate and Fe(II)-dependent dioxygenases in potato by bioinformatics, and analyzed the expression patterns of it under abiotic stress. We reviewed the manuscript that still has the following problems:
1. Line 156: the 'h' was inconsistent with the 'hour' in the abstract.
2. Lines 241-243 should be moved to the Materials and Methods section.
3. Line 261 lack of time for references.
4. Line 311,416 'hours' should be abbreviated to' h 'for the first time, using' h 'for the entire text.
5. Line 330, the reference should be removed the end of sentence, and revised in full text .
6. Line 418, 'Solanum lycopersicum' in is inconsistent with 'S. lycopersicum' in line 399. It is recommended to check the latin name of the species in full text.
7. Figure1 is unclear, and the font cannot be red, because there are red-green color blind researchers.
8. Figure 3 is unclear.

Reviewer 3 ·

Basic reporting

no comment

Experimental design

1) The CDS length and Amino acids in Table S1 do not correspond.
2) The clarity of the image is not sufficient to clearly see the content in the image, such as Figures 1,2,3,5,7.
3) The author uses blast to screen out the target genes/proteins, which cannot fully cover all genes/proteins. It is recommended that the author use the PFAM database to retrieve the target gene/protein, and then confirm it with SMART, etc.
4) Figure 3 shows an obvious error in the phylogenetic analysis.
5) There is no control in Figure 8. It is recommended to use untreated materials as the control.

Validity of the findings

no comment

·

Basic reporting

This manuscript failed to include recent literature. As an example Sonawane et al. 2022. The introduction needs to be improved by incorporating up-to-date literature.
The majority of the figures are pixilated and Fig. 3 is not readable.

Experimental design

There is a major issue with the design: the authors used a very old genome release for their study (current version DM 1-3 516 R44 - v6.1). Many revisions have been made after the mentioned genome assembly, including a complete nomenclature change. Therefore, in order to consider this manuscript for review, the authors need to redo the entire analysis using the latest version of the genome assembly.

Validity of the findings

The validity of the findings cannot be assessed since the authors used an outdated assembly as their data source.

·

Basic reporting

no comment

Experimental design

no comment

Validity of the findings

no comment

Additional comments

In this article, the authors characterized the 2-Oxoglutatrate and Fe (II)-
dependent dioxygenases (2ODD) enzyme family members in potato, by using several bioinformatic tools and the qPCR analysis. The following are the major concerns:
1. Materials and methods:
(1) The description of the stress treatment was unclear. How the plant materials were planted (In the growth bag or in the field)? how was the drought treatment conducted? The information of the salt treatment was completely lacking.

2. Results and discussion:
(1) Line 192-193, this conclusion should be more carefully made. In Table S1, the number of the proteins that were predicted to be extracellularly localized, is seemingly same with those in cytoplasmic. Additionally, why are so many 2ODD enzymes extracellularly localized? If the prediction here is reliable, the authors should cite some references which showing the subcellular locations of the proteins or their homologs listed in this table to support thieir prediction.
(2) The transcript ID of the genes in Table 1 or elsewhere in the manuscript should be changed into the formal ID of the public database (e. g. GenBank), which will facilitate the reviewer or the following researchers to retrieve the gene information or the gene sequences.
(3) In table S1, some information was seemingly incorrectly provided. For example, for “PGSC0003DMT400003153”, the amino acids were 315, but the CDS length is only 876 bp (945 bp instead?). Others should also be checked.
(4) Line 227-229, based on which critiria, these proteins were predicted to have ACO or GAOX function? Their similarity with the ACO or GAOX homologs in the other species? If so, the authors should provide the data.
(5) More detailed information was lacking in the docking results as described in line 259-273. Ten representative proteins were chosen for all the ten family groups. What are these proteins (detailed biological and molecular functions) and their substrates? According to the above description, each group seems to have diverse functions. What puzzled me here that for the docking, 2OG and ACO were used as the docking substrates for all the proteins? In group 2, most are gibberellin oxidases, as such, the GA intermediates should be used for GA20 or GA3ox, and GA1 or GA4 for GA2ox for the docking? As addition, the predicted substrate binding sites of all proteins in different groups are comprised of Y, R and S residues, which are located at the β-sheets of the protein surfaces. The situation is quite different from the previous report which well characterized the structure of the GA-binding protein, where GA clearly binds to the imbedded space-filling core (Murase et al., 2008).
(6) Line 265-266, Histidine, Histidine and Aspartate should be marked clearly in figure 5; besides, the binding sites for Fe (II) can be marked in figure 4?
(7) The expression of the genes in Figure 7 were all validated by qPCR? If not, based on which criteria that these 9 genes were selected for qPCR analysis? Meanwhile, the expression of St2odd062 was significantly up-regulated by RWT treatment in Figure 8A, whereas it was down-regulated in Figure 7C, suggesting the transcriptome data is not reliable? Same to St2ODD092.2.
(8) Line 301-309, nine genes were used for qPCR analysis of plants in response to salt and drought treatment. What are these genes? More information, and discussion of these selected genes should be provided in the result and discussion section.
(9) Information of the primer sequences were not provided for some genes in Table S3.
(10) Line 453, according to the results in figure 8A, St2ODD069 was actually not affected by drought treatment, but was only up-regulated by re-watering treatment, similar conclusions elsewhere should be more carefully made.

---

## Round 0.2 · Minor Revisions

Although the new version has much improved, the authors still need to address a few minor issues that reviewer #5 raised.

Reviewer 1 ·

Basic reporting

The authors responded to all valuable comments

Experimental design

I am generally happy with the experimental design of this study

Validity of the findings

Showing interaction effects between underlying dioxygenases and stresses are now well structured and is acceptable

Additional comments

Many thanks

·

Basic reporting

I'm satisfied with the current revision, however some in the MS still can be improved. The detailed observations and suggestions can be found in the edited PDF.

Experimental design

I'm ok with this part

Validity of the findings

I'm ok with this part

---

## Round 0.3 · Minor Revisions

I appreciate authors' effort in revising the manuscript thoroughly however the manuscript still needs editing by a language expert.

**Language Note:** The Academic Editor has identified that the English language must be improved. PeerJ can provide language editing services - please contact us at copyediting@peerj.com for pricing (be sure to provide your manuscript number and title). Alternatively, you should make your own arrangements to improve the language quality and provide details in your response letter. – PeerJ Staff

---

## Round 0.4 · accepted · Accept

The present version looks satisfactory except for a minor error. In the abstract "facilitates" should be "facilitate".